# Effects of Heavy Metal Stress on Physiology, Hydraulics, and Anatomy of Three Desert Plants in the Jinchang Mining Area, China

**DOI:** 10.3390/ijerph192315873

**Published:** 2022-11-29

**Authors:** Tianpeng Gao, Haoming Wang, Changming Li, Mingbo Zuo, Xueying Wang, Yuan Liu, Yingli Yang, Danghui Xu, Yubing Liu, Xiangwen Fang

**Affiliations:** 1School of Biological and Environmental Engineering, Xi’an University, Xi’an 710065, China; 2School of Biological and Pharmaceutical Engineering, Lanzhou Jiaotong University, Lanzhou 730070, China; 3Engineering Center for Pollution Control and Ecological Restoration in Mining of Gansu Province, Lanzhou City University, Lanzhou 730070, China; 4College of Life Sciences, Northwest Normal University, Lanzhou 730070, China; 5Institute of Environmental Health Science in Xi’an, Xi’an 710065, China; 6State Key Laboratory of Grassland Agro-Ecosystems, College of Ecology, Lanzhou University, Lanzhou 730000, China

**Keywords:** heavy metal, gas exchange, hydraulics, stem anatomy, tailings, quinoa

## Abstract

The physiological mechanisms and phytoremediation effects of three kinds of native quinoa in a desert mining area were studied. We used two different types of local soils (native soil and tailing soil) to analyze the changes in the heavy metal content, leaf physiology, photosynthetic parameters, stem hydraulics, and anatomical characteristics of potted quinoa. The results show that the chlorophyll content, photosynthetic rate, stomatal conductance, and transpiration rate of *Kochia scoparia* were decreased, but intercellular CO_2_ concentration (Ci) was increased under heavy metal stress, and the net photosynthetic rate (Pn) was decreased due to non-stomatal limitation. The gas exchange of *Chenopodium glaucum* and *Atriplex centralasiatica* showed a decrease in Pn, stomatal conductance (Gs), and transpiration rate (E) due to stomatal limitation. The three species showed a similar change in heavy metal content; they all showed elevated hydraulic parameters, decreased vessel density, and significantly thickened vessel walls under heavy metal stress. Physiological indicators such as proline content and activity of superoxide dismutase (SOD) and peroxidase (POD) increased, but the content of malondialdehyde (MDA) and glutathione (GSH), as well as catalase (CAT) activity, decreased in these three plants. Therefore, it can be concluded that these three species of quinoa, possibly the most dominant 30 desert plants in the region, showed a good adaptability and accumulation capacity under the pressure of heavy metal stress, and these plants can be good candidates for tailings remediation in the Jinchang desert mining area.

## 1. Introduction

Tailings are solid or liquid waste from mineral processing, usually stored in open sites without any protection measures, posing a major safety hazard [1,2]. Tailings contain a large amount of heavy metals, which enter the soil and groundwater through rainfall, runoff, and wind, interfering with the ecosystem and threatening the safety of human life and health [3]; therefore, the remediation of heavy metal-contaminated soil has become a global imperative [3,4,5,6].

Heavy metals in tailings can cause changes in plant physiology, hydraulics, photosynthesis and induce anatomical changes [7,8,9,10], which negatively impact the plant survival and remediation processes in contaminated environments. Understanding the enrichment mechanism of heavy metals by plants under stress of tailings can help to design the best phytoremediation scheme and improve the efficiency of phytoremediation [11,12]. Antioxidant enzyme activity and glutathione (GSH) content have changed to actively respond to oxidative damage under heavy metal stress [13]. Malondialdehyde (MDA) and proline are usually used as indicators in response to plant stress [14]. Usually, an abiotic stress-induced reduction in plant photosynthesis is mainly caused by stomatal limitation and non-stomatal limitation [15,16]. The net photosynthetic rate of plants gradually increased with the increase in heavy metal concentration in tailings, when Philippine Tung (*Reutealis trisperma*) was planted in lead and silver tailings for a short period [17]. As the proximity to the source increases and the heavy metal content increases, the stomatal conductance (Gs) and transpiration rate (E) of two *Restinga* herbaceous species—*Ipomoea imperati* and *Canavalia rosea*—are dramatically reduced [18]. Yihui et al. studied the photosynthesis of maize under Pb–Zn mineral stress and found that the net photosynthetic rate (Pn), Gs, and substomatal CO_2_ concentration (Ci) parameters increased at low concentrations but decreased at high concentration [19].

Complex heavy metals can affect the hydraulic conductivity of plants [20]. For example, high concentrations of Zn significantly reduced the root hydraulic conductivity of *Quercus suber* L. [21]. Arsenic contamination reduces root water absorption in soybean (*Glycine max* L.) [22]. Gitto et al. found that root hydraulic conductivity and water channel protein gene expression decreased with the increasing zinc concentration in hydroponically grown barley [23]. All these studies indicate that the root hydraulics decrease under heavy metal stress, while the stem hydraulics have been less studied.

Heavy metal stress also affects the anatomical structure of plants. Ahsan et al. evaluated the response of *Rosa* species (*Rosa damascena*, *Rosa bourboniana*, *Rosa Gruss an Teplitz*, and *Rosa centifolia*) after treatment with wastewater containing heavy metals. They found that the vascular bundle area of *R. centifolia* increased significantly after being treated with wastewater [24]. In contrast, when studying the effect of heavy metals on the anatomical characteristics of *Cenchrus ciliaris* L. in cement dust pollution, Al et al. found that the anatomical changes in the stems were not significant under stress and unstressed conditions, except for the size of the cells [25]. In Haji’s study, the results show that wheat root diameter and xylem vessels appeared significantly reduced under irrigation with complex heavy metal wastewater [26]. These results indicate that the changes in plant anatomy under heavy metal stress vary from species to species. Therefore, understanding changes in photosynthesis, hydraulics, and anatomy in mining plants is necessary for screening potential remediation plants.

The harsh environmental features, such as the high content of heavy metals, metallic salts, and the organic nutrient deficiencies, inhibit plant growth in tailings. These features are not conducive to vegetation restoration in desert mining areas [27,28]. At present, the most important task is to find plants that can tolerate toxicity and can grow well in tailings. The Gramineae, Chenopodiaceae, and Asteraceae families, and the genus *Tribulus* are relatively dominant species in the Jinchang mining area, Gansu province, Hexi corridor, northwestern China; they are not limited by local climatic factors. Therefore, they may be used as candidates for remediation plants. In addition, quinoa has the ability to accumulate a large number of heavy metals, which is very suitable for the remediation of heavy metal-contaminated sites [29].

Phytoremediation is less costly, sustainable, and eco-friendly than traditional physicochemical remediation [30,31], so it is a more recommended method for tailings remediation in arid and semi-arid regions because it can preserve vegetation to avoid erosion [32,33].

In this study, the contents of heavy metal, antioxidant stress index, leaf physiology, gas exchange, stem hydrodynamics, and anatomical structure were studied for three local quinoa plants: *Kochia scoparia*, *Chenopodium glaucum*, and *Atriplex centralasiatica*. The potential of these three species was evaluated as candidates for remediation plants in Jinchang tailings.

## 2. Materials and Methods

### 2.1. Soil Type and Planting Conditions

Samples were collected from 0 to 20 cm at the surface of the native soil (Xiasifen, 38°39′16.23′′ N, 102°16′46.32′′ E) and slag tailings operations (tailings dam 38°30′57.5′′ N, 102°09′4.2′′ E). The control group (virgin soil, defined as the blank group (C)) and an experimental group (T) (virgin soil mixed with tailings at a ratio of 1:1) were prepared. The whole experiment was conducted in the field experiment workstation at the engineering center of pollution control and ecological restoration of the mining area in Gansu province, which is 20 km away from the tailings dam in Jinchang city. The dominant plants were selected by survey markers in the mine area using a sampling method in October 2018. A total of three quinoa species with good growth conditions, namely, *Kochia scoparia*, *Chenopodium glaucum*, and *Atriplex centralasiatica*, were identified, and their seeds were collected from unpolluted areas near the mine site. The seedlings were planted in 60 × 60 cm seedling pots at the field rehabilitation site in Jinchang mine in May 2019 and watered thoroughly. Overall, three biological replicates were set up in each group and three plants were mixed into one biological replicate, and various indicators were measured at plant maturity after 110 days of incubation.

### 2.2. Determination of Heavy Metal Content in Soil and Plants

The soil sample was weighed at 0.2 g and put in a Teflon (TFM) cup. 6 mL of nitric acid, 2 mL of hydrofluoric acid, and 4 mL of hydrochloric acid were added and mixed with the sample; then, it was put into the microwave digestion instrument (MARS6) to start digestion. The lid of the (TFM) lysis cup was rinsed with dH_2_O three times, then 1 mL of perchloric acid was added to the Teflon cup and it was heated at 150 °C on an electric hot plate to drive the acid to a viscous state. The cup was then rinsed three times with 2 mL of ultrapure water each time, transferred to a volumetric flask, fixed to 50 mL, and diluted to determine the contents of five heavy metals by ICP-MS (PerkinElmer, NexION 300X) (PerkinElmer, Waltham, MA, USA). Each treatment had three replicates.

The plant sample was weighed at 0.2 g into the dissolution cup, 8 mL of nitric acid and 1 mL of hydrogen peroxide were added in turn. The sample was shaken gently to mix with the sample, and then left for 30 min before microwave digestion. After digestion, the sample was fixed to 50 mL and then diluted by ICP-MS to detect the concentrations of the five heavy metals.

The soil sample was weighed at 10.0 g into a 50 mL beaker and 25 mL of ultrapure water was added. The beaker was sealed with plastic wrap, stirred vigorously with a magnetic stirrer for 2 min, and left to stand for 30 min. The temperature of the sample was controlled by maintaining the water bath at 25 ± 1 °C, and the difference with the temperature of the standard buffer solution was not more than 2 °C. After calibrating the pH meter (PHS-3E), the electrode probe was immersed into the liquid surface at 1/2 the vertical depth of the suspension and the beaker was shaken gently. After the reading is stable, the pH value was recorded. The following bioconcentration factor (BCF) and transit factor (TF) are two indexes to evaluate the absorption and accumulation of heavy metals by plants as follows Equations (1) and (2).
BCF = Concentration of heavy metals in plant/Concentration of heavy metals in soil(1)
TF = Concentration of heavy metals in shoot/Concentration of heavy metals in root(2)

### 2.3. Measurement of Photosynthetic Parameters of Plants

The photosynthetic indexes of the upper and middle leaves of the plants were measured using a Li-6800 portable photosynthesizer with natural light from 9:00 to 11:00 a.m. every day as the light source during the maturation period. At least three plants were measured in one treatment, and three replicates were performed for each plant. The photosynthetic indexes measured were the net photosynthetic rate (Pn, CO_2_, μmol m^−2^ s^−1^), stomatal conductance (Gs, H_2_O, mol m^−2^ s^−1^), transpiration rate (E, H_2_O, mmol m^−2^ s^−1^), and intercellular CO_2_ concentration (Ci, CO_2_, μmol mol^−1^). After measuring the photosynthetic index, the plant leaves were quickly collected, and the chlorophyll content was measured using a spectrophotometer at 663 nm and 645 nm, respectively. Chlorophyll concentration was calculated according to the Lambert–Beer law of the following equation:A_663_ = 82.04 C_a_ + 9.27 C_b_(3)
A_645_ = 16.75 C_a_ + 45.6 C_b_(4)
where C_a_ and C_b_ are the concentrations of chlorophyll a and b; 82.04 and 9.27 are the specific absorption coefficients of chlorophyll a and b at wavelength 663 nm (Equation (3)); and 16.75 and 45.6 are the specific absorption coefficients of chlorophyll a and b at wavelength 645 nm (Equation (4)).

### 2.4. Measurement of Plant Physiological Indexes

SOD activity was assayed by measuring the ability of photochemical inhibition of nitro blue tetrazolium (NBT). Fresh leaf samples (0.2 g) were homogenized in 5 mL pre-cooled 0.05 mol/L PBS (pH 7.8) and centrifuged at 10,500× *g* 4 °C for 20 min to extract the supernatant enzyme solution. The 3 mL reaction mixture contained 50 mM potassium phosphate buffer (pH 7.8), 75 µM NBT, 13 mM L-methionine, 0.1 mM EDTA and 0.002 mM riboflavin, and 50 µL enzyme extract. Reactions were carried out at 25 °C under cool white fluorescent light for 10 min. The absorbance was read at 560 nm, and the difference of the absorbance was calculated between each sample and control. One unit of SOD was defined as the enzyme amount causing a 50% inhibition reduction in NBT, and the enzyme activity was expressed in units per mg of protein.

Activity of POD and CAT: Fresh leaf samples (0.2 g) were homogenized in 5 mL pre-cooled 0.05 mol/L PBS (pH 7.8) and centrifuged at 15,000 r/min 4 °C for 15 min to extract the supernatant enzyme solution used for POD CAT. In 3 mL of the reaction system, 0.3% H_2_O_2_ 1 mL 0.2% guaiacol 0.95 mL, 0.05 mol/L PBS (pH 7.0) 1 mL, and, finally, 0.05 mL of enzyme solution were added to start the reaction, and the rate of increase in absorbance at 470 nm was measured. Peroxidase activity was expressed in terms of changes in absorbance per mg protein per minute. Catalase activity was determined by measuring the decomposition of hydrogen peroxide. About 100 μL of enzyme extract was added into the reaction mixture containing 50 mM phosphate buffer (pH 7.0) and 20 mM H_2_O_2_. The decrease in the absorbance at 240 nm was recorded. One unit of CAT activity was defined as the amount required for decomposing 1 μmol of hydrogen peroxide/min/mg protein under assay conditions.

Measurement of MDA: The extent of membrane damage was determined by measuring MDA, which is the final byproduct of membrane lipid peroxidation. Fresh leaf samples (0.2 g) were homogenized in 5 mL of 5% trichloroacetic acid (TCA) and centrifuged at 12,000× *g* for 15 min. 2 mL supernatant was added to 2 mL of 0.6% thiobarbituric acid, mixed, and reacted in a boiling water bath for 10 min and immediately transferred to an ice bath. Then, the samples were centrifuged again. The amount of malondialdehyde was determined by measuring the absorbance of the supernatant at 450, 532, and 600 nm. The results are presented as MDA nmol L^−1^ of fresh tissue weight based on the below formula Equation (5):MDA (nmol/L FW) = [6.452 * (OD532-OD600) − 0.559 * OD450] * Vt/(Vs * W)(5)
where Vt is the extraction solution volume, Vs is the volume for measurement, and FW is the fresh weight of the sample.

Measurement of Proline: Standard curve were prepared in 0, 2, 4, 6, 8, 10 µg/mL proline standard solution; 2 mL of each standard solution was taken, and 2 mL of 3% sulfosalicylic acid, 2 mL of glacial acetic acid, and 3 mL of ninhydrin color development solution were added to the test tubes and mixed in a boiling water bath for 45 min, then removed and cooled. 5 mL of toluene was added to each tube, and the extract was shaken thoroughly and left to stratify, and the upper layer of toluene was aspirated with 0 µg/ mL proline. The standard curve of each proline was measured by colorimetry at 520 nm with 0 µg/mL proline as blank. Fresh leaf samples (0.2 g) were tanked in a test tube, adding 5 mL of 3% sulfosalicylic acid solution. Then, the tube was sealed after being boiled it for 10 min; the tube was removed and cooled to room temperature, then centrifuged at 3000 r/min for 10 min. The supernatant tanked 2 mL from the tube and color development, extraction, and colorimetry were carried out according to the standard curve method. Finally, the proline content was calculated from the standard curve.

### 2.5. Measurement of Plant Hydraulics

Branches were cut off at a distance of 3 cm from the soil surface and immediately placed in a bucket of water and covered with a black plastic bag (to prevent water loss and outside air from entering the branch through the incision) and brought to the laboratory. Before measurement, 3–5 cm of the stem was cut off at the branch incision underwater (to reduce the effect of air entering the branch during sampling), and the hydraulic conductivity (Kh, kg s^−1^ MPa^−1^) was measured using a high-pressure plant hydraulic conductivity meter (HPFM-Gen3). Each group had three biological replicates. Specific hydraulic conductivity (Ks) is calculated as follows Equation (6):Ks (kg m^−2^ s^−1^ MPa^−1^) = Kh/Ax (Ax = cross-section of stem)(6)

### 2.6. Determination of the Xylem Anatomy of the Plant Stems

The middle 3 cm of each stem was taken for anatomical study, stained using the Safranin O-Fast Green method. Paraffin sections were dewaxed to water: the sections were sequentially put into xylene I for 20 min, xylene II for 20 min, anhydrous ethanol I for 5 min, aqueous ethanol II for 5 min, and 75% alcohol for 5 min, washed with distilled water, and then stained with Safranin staining solution for 2 h. Distilled water was used to remove excess dye. Then, the slices were decolorized into 50, 70, and 80% gradient alcohol for 8 s each in turn. After decolorization, the sections were put into fast green staining solution for 20 s, dehydrated in anhydrous ethanol, and then put into clean transparent xylene for 5 min and sealed with neutral resin. Each sample was observed and photographed under the Olympus optical microscope, and the anatomical structure of each part was measured with Cas Viewer (v2.3). Each group had three biological replicates.

### 2.7. Analysis

Three biological replicates were set up for each treatment, and three plants were mixed into one biological replicate. Means, standard deviations, and standard errors were calculated and plotted for all parameters using GraphPad Prism 8. Two-way ANOVA was performed using SPSS Statistics 25, and the significance of the data between measurement groups was assessed by Tukey’s test (*p* < 0.05). One-way ANOVA was performed using SPSS for soil heavy metal content by Fisher’s least significant difference (LSD) and S-N-K (*p* < 0.05).

## 3. Results

### 3.1. pH of Soil and Heavy Metal Content in Plants and Soil

The heavy metal contents and pH value of soils are shown in Table 1, where the heavy metal contents of the primary soils were lower than the criterion (Risk Control Standards for Soil Contamination on Agricultural Land of China-GB15618-2018), which were soils uncontaminated with heavy metals. The pH of the soils in both groups was basically the same; and the soils were alkaline (pH > 7), and the Cr, Cu, Pb, and Zn in the soil of the experimental group exceeded the standard, with concentrations of 1.24, 11.41, 11.91, and 39.71 times the Chinese national secondary standard, respectively. The results are shown in Table 1.

The heavy metal pollution in the tailings mainly comes from three metals, namely, Cu, Zn, and Pb (Table 1). The corresponding heavy metal content in the three plants increased significantly after heavy metal stress (hereinafter referred to as stress). Based on the analysis results, we found that the Zn and Pb content in the leaves of the three plants increased significantly after stress compared with the control (Figure 1a,d,g). The lead content in the leaves of the three plants ranged from 1.23 to 4.77 mg/Kg in the control group; the order of the lead content in the three plants was: *Chenopodium glaucum* > *Kochia scoparia* > *Atriplex centralasiatica*. The lead content in the leaves of the three plants after stress ranged from 10.30 to 20.33 mg/Kg; the order was *Atriplex centralasiatica* > *Chenopodium glaucum* > *Kochia scoparia*. The zinc content in the leaves of the three plants was the highest; the zinc content in the leaves ranged from 10.32 to 26.91 mg/Kg in the control and from 134.8 to 269.8 mg/Kg after stress. The order of zinc content in the three plants was *Atriplex centralasiatica* > *Kochia scoparia* > *Chenopodium glaucum*. In the control, the copper content of the leaves of *Kochia scoparia* and *Chenopodium glaucum* was between 9.01 and 19.25 mg/Kg, and the copper content of the leaves of *Kochia scoparia* and *Chenopodium glaucum* increased after stress compared to the control, but the difference was not significant, while *Atriplex centralasiatica* and *Chenopodium glaucum* showed a significant difference in copper content compared to the control. Leaves showed a significantly higher copper content compared to the control and a much higher content than the leaves of the other two plants (copper content ranged from 47.32 to 60.61 mg/Kg) in the treatment groups, from high to low: *Atriplex centralasiatica > Chenopodium glaucum > Kochia scoparia*. Cr and Ni contents were the lowest in the leaves of the three plants, and Cr and Ni contents in the leaves of *Kochia scoparia* and *Atriplex centralasiatica* increased, but not significantly, after stress compared to the control, while *Chenopodium glaucum* showed a decrease in Cr and Ni contents in the leaves after stress. The heavy metal content of the leaves of the three plants after stress was, from high to low: Zn > Cu > Pb > Cr and Ni (Figure 1a,d,e).

Stems are the organ that accumulates the least amount of heavy metal elements in plant tissues. The content of zinc and lead in the stems of *Kochia scoparia* showed significant elevation, while chromium and nickel showed no significant elevation after stress, and there was no change in the copper in stems (Figure 1b). Compared to the control group, the content of Cu, Pb, and Zn increased in the stems, and Cr and Ni decreased in the leaves of *Chenopodium glaucum*, while the content of Pb and Zn increased significantly compared to the control (Figure 1e). The content of five heavy metals (Zn, Cu, Pb, Cr, and Ni) increased in the stems of *Atriplex centralasiatica* after stress, especially the content of Zn; Cu increased significantly after stress compared to the control (Figure 1h). The heavy metal contents in the stems of all three plants after stress were Zn > Cu > Pb > Cr and Ni.

Roots are important organs for the accumulation of heavy metals in plants. The content of five heavy metals (Zn, Cu, Pb, Cr, and Ni) in the roots of the three plants increased to different degrees after stress (Figure 1c,f,i); the content of Zn was significantly increased in the roots of the three plants, Pb was significantly increased in the roots of *Chenopodium glaucum* and *Kochia scoparia*, and Cu was significantly increased in the roots of *Atriplex centralasiatica*. The content of heavy metals in the roots of all three plants after stress was Zn > Cu > Pb > Cr and Ni.

*Kochia scoparia* showed different changes in the TF index for different heavy metals after being subjected to stress (Figure 2a), with elevated TF for Cr, Cu, and Pb, and decreased the TF for Ni and Zn, but none of them were significant. The TF order of the five heavy metals was Cr > Cu, Ni > Pb > Zn. *Chenopodium glaucum* showed significant decreases in the TF for three heavy metals, Cr, Ni, and Pb, after stress, while there was no change in the TF for Cu (Figure 2c). The TF order was Cr > Ni > Pb > Cu > Zn. Compared with the control, the TF of *Atriplex centralasiatica* was elevated after stress of Cu, Zn, and Pb, while the TF of Pb was significantly higher. The TF of Ni was significantly lower than that of the control after stress. After stress, the TF order for the five heavy metals was Ni > Cr > Pb > Cu > Zn (Figure 2e).

Compared with the control, the treated group of *Kochia scoparia* showed a significant decrease in BCF for Cr, Cu, Zn, and Pb; the BCF order for the five heavy metals was Zn > Ni > Cu > Cr > Pb (Figure 2b). *Chenopodium glaucum* showed a significant decrease in BCF of Pb, Zn, Cu, and Ni after stress; the BCF order for the five heavy metals was Zn > Ni > Cu > Cr > Pb (Figure 2d). After stress, *Atriplex centralasiatica* showed a significant decrease in BCF for Cu and Pb, and a decrease for Zn, Ni, and Cr, compared to the control, but it was not significant. The BCF order for the five heavy metals was Zn > Cu > Ni > Cr > Pb (Figure 2f).

### 3.2. Effects of Heavy Metals on Plant Photosynthesis

The Pn, Gs, and E of the three plants showed significant decreases after stress compared with the control, and interplant comparisons showed that *Atriplex centralasiatica* was significantly higher than *Chenopodium glaucum*, while *Chenopodium glaucum* was significantly higher than *Kochia scoparia* (Figure 3a,c,d). Ci is an important basis for determining whether the photosynthetic rate of plants is affected by stomatal factors. After stress, *Kochia scoparia* showed a significant increase in Ci under tailings stress, while *Atriplex centralasiatica* and *Chenopodium glaucum* showed a significant decrease compared with the control, and interplant comparisons showed that the Ci value of *Atriplex centralasiatica* had significantly decreased compared to the other two plants, *Kochia scoparia* was not significant from *Chenopodium glaucum* (Figure 3b). The chlorophyll content of the three plants also showed different degrees of decrease after stress; *Chenopodium glaucum* showed a significant decrease in Chl a+b after stress compared to the control, while *Atriplex centralasiatica* and *Kochia scoparia* showed a decrease but it was not significant. The interplant comparison showed that *Atriplex centralasiatica* had the lowest Chl a+b and a significant difference with the other two plants (Figure 3e). Plant height is a visual indicator of how well plants adapt to stress. The plant height of the three plants showed different degrees of reduction after stress. The plant height of *Kochia scoparia* and *Chenopodium glaucum* showed a significant decrease after stress compared to the control, while the height of *Atriplex centralasiatica* showed a decrease but it was not significant (Figure 3f).

### 3.3. Effects of Heavy Metal Stress on Plant Physiology

In the control there was no significant difference in the SOD activity of the three plants (Figure 4a); *Kochia scoparia* had higher SOD activity than that of *Chenopodium glaucum*, while the SOD activity of *Chenopodium glaucum* was higher than that of *Atriplex centralasiatica*. The three plants showed similar trend of higher activity than the control under heavy metal stress (Figure 4a); the SOD activity of *Atriplex centralasiatica* was higher than that of the other two plants and significantly higher than the control. The interplant comparison showed that the SOD activity of *Kochia scoparia* was greater than that of *Chenopodium glaucum*.

The POD activity of *Kochia scoparia* was significantly higher than that of the other two plants and the control, while the POD activity of *Atriplex centralasiatica* was again higher than that of *Chenopodium glaucum* and there was no significant difference. The POD changes in the three plants after stress showed the same trend; the POD activity of *Chenopodium glaucum* and *Atriplex centralasiatica* were both significantly higher than the control, and the differences among plants showed that *Kochia scoparia* was higher than *Atriplex centralasiatica*. *Chenopodium glaucum* had a significantly lower POD activity than the other two plants (Figure 4b). It was found that the CAT activity of the plants in both the control and treatment groups decreased in a gradient from left to right (Figure 4c), with a significant difference between the maximum and minimum in the control, and a decreasing trend in CAT activity of all three plants after stress compared to the control, but it was not significant.

The concentration of MDA and GSH showed no significantly different change among the three plants in the control group. The three plants showed a similar trend after heavy metal stress. The MDA concentration of all three plants was significantly lower after stress than the control. The GSH content of *Kochia scoparia* and *Atriplex centralasiatica* was significantly lower than the control, while the GSH content of *Chenopodium glaucum* was lower than the control but the difference was not significant. The difference between plants after stress was not significant (Figure 4d,e).

The proline content showed no significant difference among the three plants in the control. After stress, the proline content of the three plants showed a higher trend compared to the control, and that of *Chenopodium glaucum* was significantly higher than the control. The difference between plants showed that *Chenopodium glaucum* was significantly higher than the other two plants (Figure 4f).

### 3.4. Effects of Heavy Metals on the Hydraulics and Anatomy of Plant Stems

The hydraulic conductivity of the stems of *Atriplex centralasiatica* was significantly higher than the control under stress, while that of *Kochia scoparia* and *Chenopodium glaucum* showed no significant difference. The interplant comparison showed that the hydraulic conductivity of the stems of *Atriplex centralasiatica* was much higher and significantly different from the other two plants (Figure 5e). The Ks results of the three plants showed an increase in specific conductance after stress, and the interplant comparison showed that the specific conductance of *Atriplex centralasiatica* was higher than the other two plants but not significantly (Figure 5f). The changes in plant anatomy can directly reflect the degree of stress tolerance of plants under stress, and the vessel density and relative sparing area of the three plants were significantly reduced after stress. A comparison among the plants showed that the order of vessel density was *Kochia scoparia* > *Chenopodium glaucum* > *Atriplex centralasiatica* after stress, and the order of relative sparing area was *Kochia scoparia* > *Atriplex centralasiatica* > *Chenopodium glaucum* (Figure 5b,d and Figure 6).

The vessel diameter of *Atriplex centralasiatica* and *Chenopodium glaucum* did not increase significantly after stress, and that of *Kochia scoparia* did not decrease significantly after stress. It is shown that the vessel diameter of *Atriplex centralasiatica* was significantly higher than that of the others by interplant comparison. The vessel diameter of *Atriplex centralasiatica* was significantly higher than that of the other two species (Figure 5a). The thickness of the vessel walls of the three plants had different increases after stress. For *Atriplex centralasiatica* and *Chenopodium glaucum*, the thickness of the vessel walls was significantly higher than the control, that of *Kochia scoparia* was higher than the control but not significant, and the interplant comparison showed that *Atriplex centralasiatica* had significantly thicker vessel walls than the other two plants (Figure 5c and Figure 6).

The morphology of plant stems and the distribution characteristics of vessels vary with plant species, which largely determines the hydraulic conductivity and water transport regulation of plant stems. Those are important anatomical features that determine plant water consumption (Figure 6)

## 4. Discussion

This study indicated that the contents of Cu, Cr, Ni, Pb, and Zn in the area of weak alkaline Pb–Zn tailings in Jinchang were higher than the local background values. The heavy metal contents of Cr Pb, Zn, and Cu in the tailings are toxic to plants (Table 1). Figure 2 shows that none of the plants’ heavy metal concentrations were >1000 mg/kg in the above-ground parts of the plants and none of them were hyperaccumulator plants [34]. However, the tolerance and accumulation capacity of heavy metals may contribute to plant stability [35]. The three plants showed the same trend of enrichment under tailings stress, both with the highest Zn content, followed by the Cu and Pb content (Figure 1), which may be explained by the fact that Zn and Cu are essential elements for plant growth, while Pb is a non-essential element [36,37]. The possible reason for more Pb in plant tissues than Cr and Ni, which are non-essential elements, is that Pb content is highest in the lead–zinc tailings, and plants cannot avoid taking up Pb when they are stressed [38]. The order of magnitude of heavy metals in the three plants after stress was *Atriplex centralasiatic* > *Kochia scoparia* > *Chenopodium glaucum*.

Higher TF values indicated that plants can transport and distribute more heavy metals [39]. The TF magnitude of the five heavy metals in *Kochia scopari* after stress in this study was Cr > Cu, Ni > Pb > Zn (Figure 2a). The TF of *Chenopodium glaucum* to five heavy metals after stress was, from high to low: Cr > Ni > Pb > Cu > Zn (Figure 2c). These two plants had the highest TF for Cr, implying that they can be selected as potential candidate remediation plants for mining areas with high Cr content in the soil. In addition, the TF order of *Atriplex centralasiatic* for the five heavy metal is Ni > Cr > Pb > Cu > Zn (Figure 2e); therefore, it can be selected as a potential candidate remediation plant for the phytoremediation of nickel-mining areas. All three plants have a higher TF than 1.0 for the five heavy metals, which indicates that all three plants can effectively transfer heavy metals to the above-ground parts and facilitate the bioextraction of soil heavy metals. The BCF can be used to assess the potential for phytoremediation [40]. From Figure 2 it is clear that none of the three plants had a BCF for the five heavy metals greater than 1.0. Among the plants suitable for phytostabilization, a BCF < 1 indicates that a particular species is unable to extract significant amounts of metals from the soil [41], indicating that all three plants are expected to be potential candidates for phytostabilization. The BCF value of the blank group was significantly higher than that of the treated group, which was caused by the fact that the background value of soil heavy metals in the treated group was much higher than that of the blank group.

The results show that all three plants under heavy metal stress in tailings showed different degrees of decline in plant height compared with the control (Figure 3f); the decrease of *Kochia scoparia* and *Chenopodium glaucum* was significant compared with the control, and that of *Atriplex centralasiatic* was not significant. This may be due to the high content of heavy metals in the soil, which limited the uptake of nutrients by the plants and thus leads to a decrease in plant growth [42]. This suggests that *Atriplex centralasiatic* is better adapted to heavy metals than the other two plants. The change in plant growth under complex heavy metal stress may also be related to plant photosynthesis [43]. The chlorophyll content of the three plants after stress of complex heavy metals in the tailings showed different degrees of decrease compared to the control (Figure 3e), which may be due to the fact that heavy metal stress leads to the disruption of the chloroplast structure and the disruption of the electron transport chain [44]. The high concentration of Zn in tailings may also contribute to the decrease in chlorophyll, and although Zn is an essential cofactor for enzymes involved in carbohydrate metabolism and other physiological processes [45], Zn is detrimental at high levels because Zn can displace the Mg cofactor in chlorophyll, thereby disrupting photosynthesis, limiting growth, and causing plants to fade to green [26]. For gas exchange parameters, it has been shown that the significant decrease in the net photosynthetic rate of plants under heavy metal stress is related to stomatal restriction [46,47,48]. We found that *Atriplex centralasiatic* and *Chenopodium glaucum* showed a significant decrease in Pn along with a decrease in Gs and E after treatment, and, under heavy metal stress, Ci values also showed significant decreases; these results suggest that the heavy metal stress-induced Pn reduction in *Atriplex centralasiatic* and *Chenopodium glaucum* may be related to stomatal restriction, and the results of Saradadevi et al. in their study on wheat support this speculation [49]. Mariz et al. found that stomatal restriction resulted in decreased Pn and disordered glucose metabolism in *Lactuca sativa* L. under Al stress [50]. Guo et al. studied the photosynthetic changes of Cd-tolerant wheat (*Triticum aestivum*) under Cd stress and found that Cd stress decreased the Ci, Pn, and Gs values of wheat [51]. This all suggests that the direct effect of complex heavy metal stress on stomata may be one of the reasons for the decrease in Pn.

*Kochia scoparia* had a different Ci after heavy metal stress than the other two plants; our results show that a significant decrease in Pn in *Kochia scoparia* after stress was not due to a lower Ci in the leaves, as the Ci values in the leaves of these plants were even significantly higher than that in the control leaves (Figure 3b). The high Ci could be explained by a low Pn and/or increased dark respiration rate, which was frequently detected in heavy metal-treated plants [52]. Therefore, the significant reduction in Pn in *Kochia scoparia* leaves after stress indicates that the direct effect of complex heavy metals on stomatal closure was not responsible for the significant reduction in Pn, and the reduction in Pn in *Kochia scoparia* was a non-stomatal limiting factor. Similar results show that the willow seedlings (*Salix alba* L.) also had a decrease in Pn and Gs and an increase in Ci under the compound heavy metal stress of Cu, Pb, and Cd [53]. Zhou et al., in their study of Sb stress on *Acorus calamus*, also found a non-stomatal limitation of plant Pn decrease and Ci increase under heavy metal stress [54]. This all indicates that non-stomatal-limited Ci increase has an important role in Pn reduction. In summary, the reasons for Pn reduction in plants vary from species to species.

Plants produce excess ROS under complex heavy metal stress [55,56], which can rapidly damage biomolecular structures (DNA, RNA, and proteins) and membranes through lipid peroxidation [57], leading to metabolic disorders and cell death in plants [58]. SOD, as the main antioxidant enzyme, can be the first to balance the excess intracellular ROS. The SOD activity of three desert plants increased after heavy metal stress and was able to convert superoxide radical (O_2_^•−^), which shows good adaptation to stress, but the conversion product of SOD hydrogen peroxide (H_2_O_2_) is still toxic and eliminated by subsequent POD (Figure 5a) [56]. CAT can be involved in the conversion to H_2_O [59]. Figure 5b,c shows that POD activity increased and CAT activity decreased in the three plants after heavy metal stress compared to the control. CAT plays an important role in reducing plant H_2_O_2_ content and mitigates phytotoxicity by directly decomposing H_2_O_2_ [60], but it is likely that excess production of ROS by heavy metal stress can inactivate CAT activity at higher concentrations of heavy metals, probably by inactivating the enzyme bound to the heme group [61]. We hypothesized that the decrease in the CAT activity of plants after heavy metal stress is compensated by the increase in POD activity to scavenge hydrogen peroxide (H_2_O_2_). Plants under heavy metal stress can be used as an indicator of oxidative stress by measuring MDA content [62]. The experimental results show a decrease in leaf MDA content in the three desert plants under heavy metal stress, probably due to an increase in antioxidant enzyme activity, thus reducing H_2_O_2_ levels and membrane damage (Figure 4d). Glutathione is the most abundant non-protein thiol in cells. Besides being involved in the synthesis of phytochelatins, it is involved in heavy metal tolerance in several ways, including sequestering toxic heavy metal ions from the cytosol and neutralizing heavy metal-induced reactive oxygen species through the ascorbate–glutathione cycle [63].The GSH content of plants after stress was significantly lower than that of the control (Figure 4e), which may be due to the fact that heavy metal-threatened plants require constant consumption of GSH for detoxification. Proline, a stress-responsive amino acid, is produced by plants in response to adversity and plays a protective role against biomolecules such as lipids and lipoproteins [64]. Our results show that the proline content in the leaves of all three plant after stress was higher than that of the control (Figure 4f), which is consistent with the finding of Gajewska et al. that the proline content in plants is positively correlated with the concentration and duration of exposure to heavy metals.

Prolonged exposure to heavy metals alters the xylem structure, as well as the vessel characteristics of plants, and these altered anatomical features are the basis of their physiological adaptation to heavy metal stress [65,66]. The xylem vessel diameter of *Atriplex centralasiatic* and *Chenopodium glaucum* under heavy metal stress showed a significant increase compared to the control, while *Kochia scoparia* showed a decrease (Figure 5a), and the vessel density of all three plants showed a significant decrease compared with the control (Figure 5b). This is consistent with the results of Raju et al. in their study on the effects of heavy metals on *Avicenna marina*, who also found that the density of the xylem vessels of plant stems decreased under heavy metal stress [67]. Samad et al., in their study of chickpea (*Cicer arietinum* L.) in response to aluminum stress, also found that aluminum stress reduced xylem vessels [68]. In this study, we observed such changes in *Atriplex centralasiatic* and *Chenopodium glaucum*, and we speculated that, although heavy metal stress decreased the density of vessels in the xylem of plant stems, plants counteract it by expanding the vessel diameter or area, which may be a positive expression of plant resistance to stress. Plants have different response mechanisms to heavy metal stress, which can detoxify heavy metals by chelating them through the cell wall [57,69]. Mohajjel et al. found that low concentrations of heavy metals can increase wall thickness by inducing lignin deposition [70]. In the present study, the changes in the stem anatomical structure of the three plants were consistent with the translocation of heavy metals to the cell wall. Both the xylem vessel walls of the three plants showed thickening in contrast to the control under heavy metal stress. Cavitation induced by heavy metal results from air seeding related to the mechanical properties of pit membranes via bordered pits’ vulnerability to air seeding, which damages these membranes and increases the risk [71]. In order to prevent implosion, xylem conduits need to be reinforced and have lignified cell walls [72,73]. This also explains the phenomenon of the thickening of plant vessel walls under heavy metal stress.

The changes in plant hydraulics under heavy metal stress can be influenced by several factors [20], the anatomical characteristics of the stem have a great influence on the hydraulic efficiency and bolus propagation resistance of the stem [74], and the increase in Kh and Ks after stress in the three plants may be caused by the expansion of vessel diameter [75]. It has also been shown that the reduction in vessel density decreased the hydraulics of plants [76]. In plants without lethal toxic symptoms, the content of heavy metal ions rarely reaches a solution concentration sufficient to cause osmotic disorder. Possible causes of water uptake by plants are indirectly regulated by changes in endogenous factors (e.g., root anatomy and/or morphology) [77]. Excess heavy metal stress also affects the efficiency of water flow by reducing the transpiration rate and/or by altering the stomatal resistance of leaves [78]. The results show that the heavy metals of tailings caused a decrease in E and Gs in the three plants, and the three plants showed an increase in hydraulics after stress, which is consistent with Adriana et al. who studied changes in *Salix purpurea* L. under heavy metal stress by expanding the vessel diameter to maintain hydraulics [75]. The stomatal conductance, transpiration rate, vessel density, and relative sparing area of three plants decreased after stress compared with the control in the present study, but the vascular diameter, Ks, and Kh of the plant hydraulic parameters have increased. Based on the above experiments, the change in valve stem hydraulics under heavy metal stress is mainly related to the anatomical structure of valve stem, although many experiments show that there is a close relationship between Gs, E, and stem hydraulics [79,80].

## 5. Conclusions

Heavy metal stress in tailings resulted in variations in the plant height and anatomical characteristics of the three plants, with a decrease in plant height, xylem vessel density, and relative sparing area, as well as an increase in vessel diameter. Physiological indicators, proline content and the activity of SOD and POD, increased, but the content of MDA and the activity of GSH and CAT decreased in these three plants. The photosynthetic parameters, such as Pn, E, and Gs, of the three plants were significantly reduced under heavy metal stress. Plant stem hydraulic conductivity and specific conductance decreased, which may be caused by changes in the plant stem structure and transpiration rate. The total heavy metal levels (contents) of the three plants increased; although none of them reached the level of super-enriched plants, they can be used as heavy metal-enriched plants. For these five heavy metals (Cr, Ni, Cu, Pb, and Zn), TF > 1 and BCF < 1 of these three plants, and these three plants are the local dominant species in Jinchang desert, which can be used as candidate plants for phytoremediation. The study found that the Kh, vessel wall thickness, and Ks of *Atriplex centralasiatica* were significantly increased in the treatment group, which means that the plant could transfer the heavy metals absorbed by the roots to the above-ground part more effectively, and *Atriplex centralasiatica* was considered as the most promising candidate remediation plant for Zn contamination because of its strong Zn enrichment ability.

## Figures and Tables

**Figure 1 ijerph-19-15873-f001:**
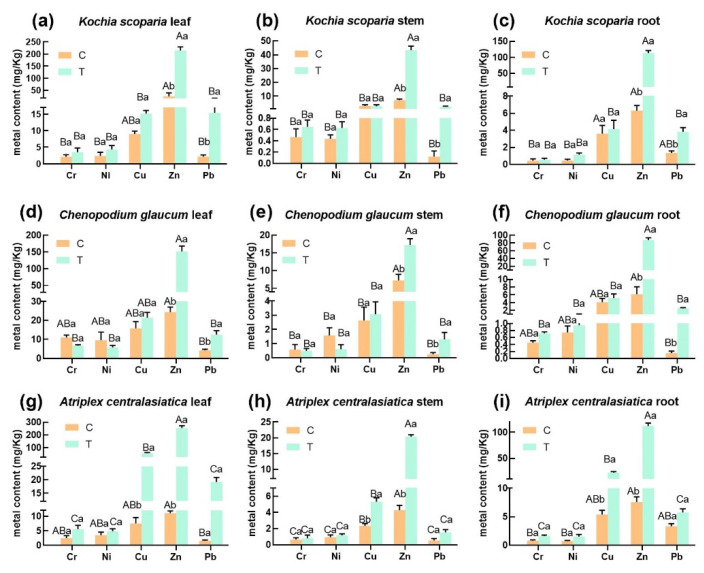
Five heavy metal contents in leaves, stems, and roots of three plant species. Bars indicate mean ± SD (*n* = 3). Different capital letters indicate significant differences between groups (between different plants) and different lowercase letters indicate significant differences within groups (control and treatment of the same plant) using the Tukey test (*p* < 0.05). C represents the control group; T represents the experimental group.

**Figure 2 ijerph-19-15873-f002:**
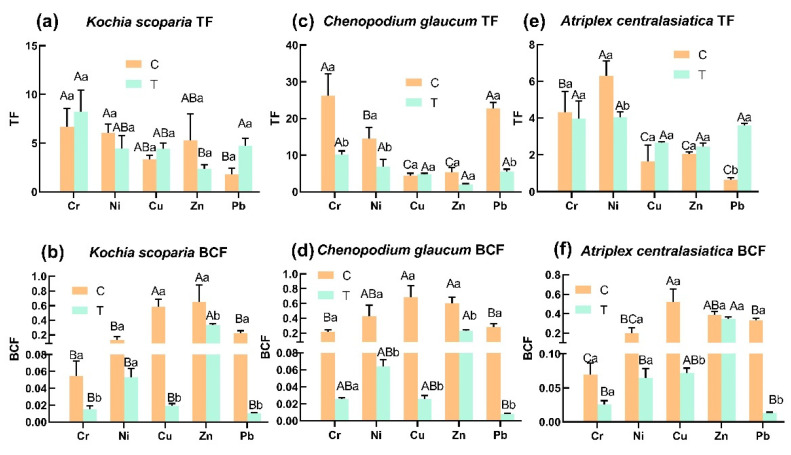
Enrichment of five heavy metals in three plant species. Bars indicate mean ± SD (*n* = 3). Different capital letters indicated significant differences between groups (between different plants) and different lowercase letters indicate significant differences within groups (control and treatment of the same plant) using the Tukey test (*p* < 0.05). C represents the control group; T represents the experimental group.

**Figure 3 ijerph-19-15873-f003:**
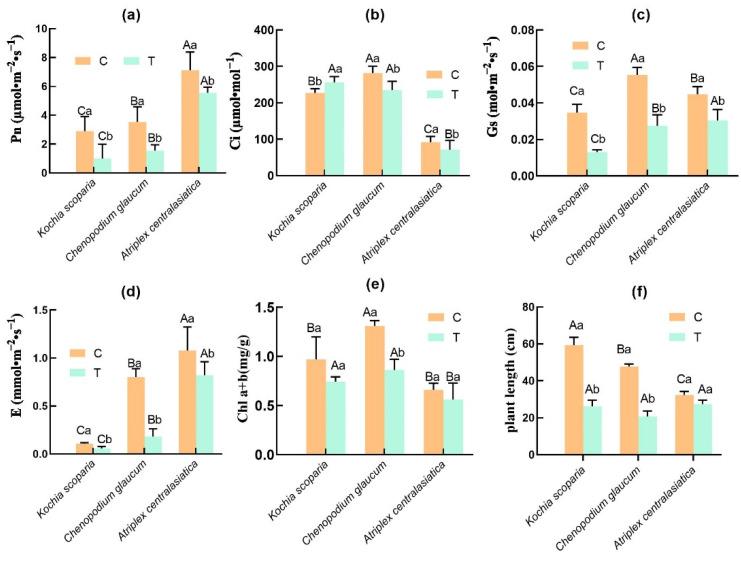
Changes in plant gas exchange parameters and plant height after tailings stress. Bars indicate mean ± SD (*n* = 3). Different capital letters indicate significant differences between groups (between different plants) and different lowercase letters indicate significant differences within groups (control and treatment of the same plant) using the Tukey test (*p* < 0.05). C represents the control group; T represents the experimental group. (**a**) Effects of heavy metals on the Pn of the three plants. (**b**) Effects of heavy metals on the Ci of the three plants. (**c**) Effects of heavy metals on the Gs of the three plants. (**d**) Effects of heavy metals on the E of the three plants. (**e**) Effects of heavy metals on the Chl a+b of the three plants. (**f**) Effects of heavy metals on the Plant length of the three plants.

**Figure 4 ijerph-19-15873-f004:**
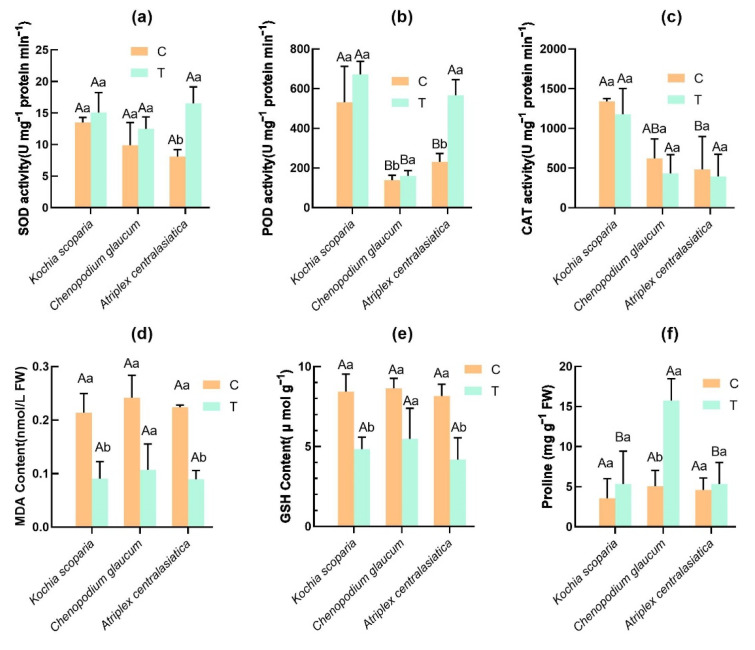
Physiological changes in leaves of three plants after tailing stress. Bars indicate mean ± SD (*n* = 3). Different capital letters indicate significant differences between groups (between different plants) and different lowercase letters indicate significant differences within groups (control and treatment of the same plant) using the Tukey test (*p* < 0.05). C represents the control group; T represents the experimental group. (**a**) Effects of heavy metal stress on SOD activity of the three plants. (**b**) Effects of heavy metal stress on POD activity of the three plants. (**c**) Effects of heavy metal stress on CAT activity of the three plants. (**d**) Effects of heavy metal stress on MDA content of the three plants. (**e**) Effects of heavy metal stress on GSH content of the three plants. (**f**) Effects of heavy metal stress on Proline of the three plants.

**Figure 5 ijerph-19-15873-f005:**
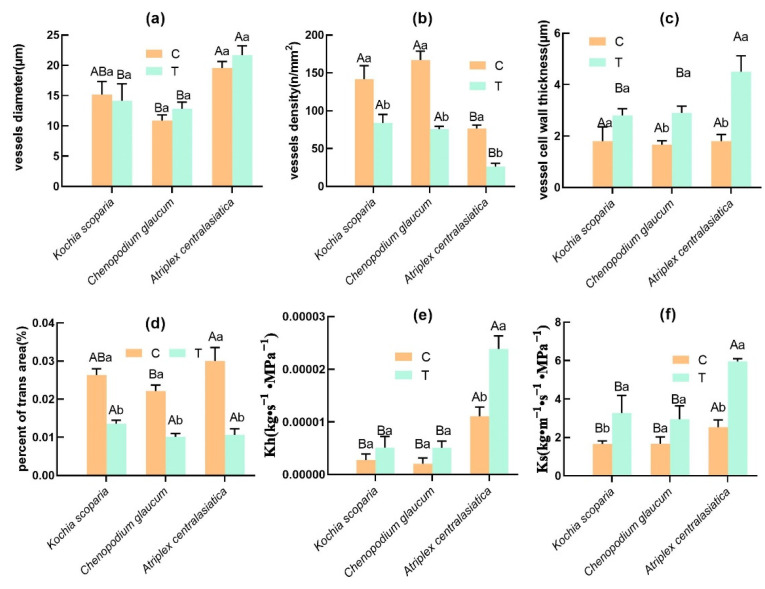
Changes in hydraulic and anatomical parameters of three plant species under tailings stress. Bars indicate mean ± SD (*n* = 3). Different capital letters indicate significant differences between groups (between different plants) and different lowercase letters indicate significant differences within groups (control and treatment of the same plant) using the Tukey test (*p* < 0.05). C represents the control group; T represents the experimental group. (**a**) Effects of heavy metals on vessels diameter of the three plants. (**b**) Effects of heavy metals on vessels density of the three plants. (**c**) Effects of heavy metals on vessels cell wall thickness of the three plants. (**d**) Effects of heavy metals on percent of trans area of the three plants. (**e**) Effects of heavy metals on Kh of the three plants. (**f**) Effects of heavy metals on Ks of the three plants.

**Figure 6 ijerph-19-15873-f006:**
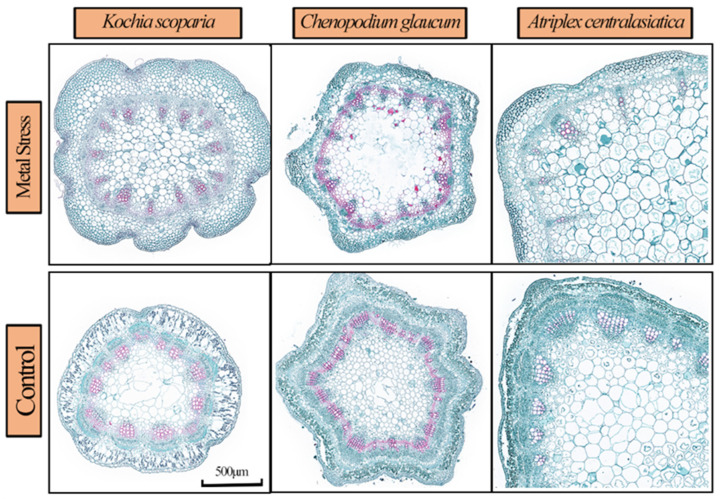
Stem cross-sections of three plants under native soil and tailings stress conditions.

**Table 1 ijerph-19-15873-t001:** Heavy metal content and Chinese National Standards.

Soil	pH	Cr (mg/Kg)	Ni (mg/Kg)	Cu (mg/Kg)	Zn (mg/Kg)	Pb (mg/Kg)
CK	8.45	55.7 ± 4.51	25.6 ± 3.57	26.7 ± 3.52	59.3 ± 6.50	16.3 ± 0.92
T	8.23	312 ± 16.9 **	114 ± 9.43 *	1141 ± 17.5 **	1113 ± 26.2 **	2025 ± 53.1 **
Standard	>7.5	250	190	100	300	170

(0–20 cm depth of two soil types, C and T; * indicates significant difference compared with control, *p* < 0.05. ** indicates highly significant difference compared with control, *p* < 0.01. C represents the control group; T represents the experimental group).

## Data Availability

The data presented in this study are available on request from the corresponding author.

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
