# Peer review of "Effects of Heavy Metal Stress on Physiology, Hydraulics, and Anatomy of Three Desert Plants in the Jinchang Mining Area, China"

_ijerph, 2022, doi:10.3390/ijerph192315873_

Round 1

Reviewer 1 Report

Readability needs improvement. In addition, an English review is strongly recommendable. Several parts are confusing. Several parts are confused. Many sentences are subjectless especially in the methodology. The results present argumentative excerpts. Results should objectively describe the data obtained and not discuss them.

Some abbreviations do not correspond to their terminologies: transpiration rate (E), glutathione (GSH), and virgin soil (CK). I recommend abbreviations related to the words they refer to. On the other hand, some abbreviations have no explanation: BCF, and TF (the formula is shown but the meaning is not), LSD.

Several symbols and abbreviations are not explained in the graphs. Graphs should be self-explanatory. Legends must contain all abbreviations and symbols used in each graph. All figures containing graphs need corrections in the legends.

Several species are without italics and their taxonomic authorities are missing. Taxonomic authority is required for the first citation of a species. The species first citation is with the genus in full; after that, the genus is abbreviated, unless there is a risk of misunderstanding. You need to review the entire manuscript in view of these aspects.

The are no citations and references for almost all methodological procedures. In the anatomical procedure, there are some mistakes, e.g. “fannin” dye. “After decolorization, the sections were put into solid solution for 20s”, line 210 à How is it possible? What is this green dye? I think you may have used the safranin and fast green dyes because they are the most commonly used in plant anatomy. How many plants (repetitions) were analyzed in each treatment in the anatomical approach?

Why was the chlorophyll content measured by spectrophotometry at 663 nm for middle leaves and 665 nm for upper leaves? Line 145.

In the Results, sometimes the statistical analysis was not followed, e.g. line 263 ("nickel also showed elevation but not significant after stress").

Reviewer 2 Report

Manuscript ID: ijerph-1994541

Title: Effects of heavy metal stress on physiology, hydraulics, and anatomy of three desert plants in the Jinchang mining area, China

The manuscript describes the evaluation of the effect of exposure to mine tailing residues on three desert plant species (Kochia scoparia, Chenopodium glaucum, and Atriplex centralasiatica), different parameters were evaluated to establish the effects related to the bioaccumulation and translocation of five Heavy Metals (Cr, Ni, Cu, Zn, and Pb) in plant tissues, such as leaf physiology, photosynthetic parameters, hydraulic and anatomical of the stem. Experimental data give relevant information about the heavy metal accumulation capacity of the three evaluated plant species, the physiological response, and the resistance mechanisms implicated in counteracting heavy metal toxicity. The authors propose the three plants evaluated with high potential for phytoremediation of heavy metal-polluted soils in desert environments.

In general, the whole manuscript must be reviewed to correct the format, there are several format mistakes in all sections.

I suggest to prior to accepting the manuscript for publication in IJERPH, the following recommendations must be addressed.

Abstract

Lines 17-18, mention the three plants species evaluated

Lines 20-25, indicate acronyms of the evaluated parameter first time mentioned

Line 24, showed a decrease, instead of just decrease

Line 24, the names of the plant species must be in italics

Lines 27-30, add a space between the enzyme or metabolite names and the parenthesis with acronyms

Line 44, in “…and anatomical changes…”, maybe use “and induce anatomical changes”

Introduction

Lines 45-47, in the fragment “Understanding the mechanism of plants under heavy metal stress of tailings can design the best phytoremediation scheme and improve the efficiency of phytoremediation”, the main idea is not clear, which kind of mechanism intended to determinate, please describe.

Line 46, use “can help to”, instead of just “can”

Line 52, maybe “short” instead of “shore”

Materials and Methods

Line 117, add a space in “0.2g”

Line 118, use mL instead of ml

Line 126, add a space in “1mL”

Line 133, change “PH meter” to “pH meter”

Line 146, please explain in the text why to make chlorophyll at two wavelengths so close

Lines 150-154, add spaces between numerical values and units

Lines 159-1170, add spaces between numerical values and units

Lines 162 and 167, in the peroxide formula use subscripts for numbers

Line 171, use the full name malondialdehyde (MDA), and then just acronym MDA

In centrifugation parameters there are several formats (i.e. 10500g 4 for 20min; 15000r/min 4 for 15 min; 12000 × g for 15 min; 3000r/min 190 for 10min), please use the same in all methodological descriptions

Line 178, in …the supernatant at 532 nm 450nm 600nm… please review which is the correct wavelength, or correct to “450, 532, and 600 nm”

Line 178, in “MDA nmol L-1”, use superscript for the number “-1”

Line 182, review and correct numeric values in “Proline :After preparing 0.2.4.6.8.10µg/ml proline standard solution”

Lines 187-191, add spaces between numerical values and units

Line 208, could be “Fannin staining solution”

Line 209, could be “into 50, 70 and 80%” instead of “into 50%, 70% and 80%

Lines 210-212, add spaces between numerical values and units

Line 214, add information about version and supplier of Cas Viewer

Lines 220 and 221, change “P” for “p” in italics

Lines 225 and 226, eliminate extra spaces in the quotes

In table 1, add spaces between the chemical element and parenthesis

Line 234, use p in italics, use uppercase in “indicates”, and eliminate the extra period in “p < 0.01.).”

Lines 240-260, use mg/Kg instead of mg/kg, all the bacterial species names must be in italics

Lines 265-275, all the bacterial species names must be in italics

In figure 1, all the bacterial species names must be in italics

Line 283, use p in italics

In figure 2, suggest to first show BCF graphics and subsequently the TF graphics, as well all the bacterial species names must be in italics

Line 306, use p in italics

Lines 307-326, all the bacterial species names must be in italics

Line 331, use p in italics

Figures 3, 4 and 5, all the bacterial species names must be in italics

Lines 380-382, all the bacterial species names must be in italics

Lines 388-397, all the bacterial species names must be in italics

Discuss why the BCF values were higher in control experiments. 

Conclusions

Please address the following question in the conclusions

According to the findings of the study which of the three species evaluated have a higher potential for phytoremediation?, how do changes in length and anatomical characteristics affect the performance of the evaluated species in phytoremediation?, according to the results which is the main heavy metal extracted?, also if possible include information about future trends for research or possible application approaches.

Round 2

Reviewer 2 Report

All the commentaries in the manuscript were addressed by the authors